# Changes in Little Cigar and Cigarillo Use during the COVID-19 Pandemic: A Cross-Sectional Analysis of a Nationally Representative Sample of Young Adult Users

**DOI:** 10.3390/ijerph19158933

**Published:** 2022-07-22

**Authors:** Eugenia Lee, Stephanie Pike Moore, Erika Trapl, Craig S. Fryer, Douglas Gunzler, Kymberle L. Sterling

**Affiliations:** 1Department of Health Promotion and Behavioral Sciences, University of Texas Health Science Center at Houston, Dallas Regional Campus, Dallas, TX 75207, USA; kymberle.l.sterling@uth.tmc.edu; 2Population and Quantitative Health Sciences, Case Western Reserve University, Cleveland, OH 44106, USA; snp39@case.edu (S.P.M.); erika.trapl@case.edu (E.T.); dgunzler@metrohealth.org (D.G.); 3Department of Behavioral and Community Health, School of Public Health, University of Maryland, College Park, MD 20742, USA; csfryer@umd.edu

**Keywords:** little cigar, cigarillo, risk perceptions, COVID-19, pandemic, cessation

## Abstract

We examined the smoking behaviors of U.S. young adults ages 18–36 regarding little cigars and cigarillos (LCCs) during the COVID-19 pandemic. Survey data were collected from a nationally representative sample of young adults between October and November 2020. Respondents who reported using LCCs with tobacco (CAI) and/or with marijuana (CAB) within the past 6 months prior to the survey (*n* = 399) were included in the study. Logistic regression analyses assessed the association between their perceived risk of having COVID when smoking LCCs and pandemic-related behavioral changes in CAI and CAB use (e.g., worrying, quit attempts, smoking more, smoking less). Findings showed that users with a higher perceived risk of getting COVID-19 when smoking LCCs were more likely to endorse trying to quit CAI and CAB during the pandemic. Compared to the non-Hispanic White population, the non-Hispanic Black population were less likely to endorse smoking less CAI and trying to quit CAB during the pandemic. Dual users of CAI and CAB and females were more likely to endorse smoking more CAB compared to CAB-only users and males, respectively. Tailored cessation strategies are needed for dual users, non-Hispanic Black young adults, and young women. Raising awareness about the risks of LCC use can be an effective strategy for LCC smoking cessation.

## 1. Introduction

The COVID-19 pandemic led to significant changes in lifestyles (e.g., virtual learning, social distancing, and remote working) and economic instability, which resulted in stress, anxiety, and isolation for many people. As the association between negative affect and smoking suggests [1,2], people tend to increase tobacco consumption following the exposure to stressful events, such as natural and human-made disasters [3,4,5,6]. The COVID-19 pandemic, however, is different from those disasters because of its direct and nationwide impact on the health of all individuals. While long-term effects of the COVID-19 pandemic on tobacco use are still being examined, changes in smoking and quitting behaviors were observed during the early phases of the pandemic. One study found that 28% of users of cigarettes, e-cigarettes, cigars, pipes, snus/snuff/dip/chew, hookah/waterpipe, or dissolvable (*n* = 291) increased cigarette use during the pandemic, while 15% decreased tobacco use and 24.5% made a quit attempt [7]. In another study of past 30 day cigar users of traditional large cigars, little cigars, or cigarillos (*n* = 777), 40.9% of participants reported increased tobacco use compared to 17.8% who reported decreased use; 46.5% had made a quit attempt since the start of the pandemic [8]. Reasons for increased tobacco use included pandemic-induced stress, anxiety, and boredom [7,9,10]. On the other hand, schedule changes, being around non-smokers, including children, and COVID-19-related health risks accounted for decreased tobacco use [7,9,10,11].

While more data exists on other tobacco products, the use of little cigars and cigarillos (LCCs) during the COVID-19 pandemic has been understudied. LCCs are popular, particularly among young adults [12,13,14], males [14,15], and people of color [12,14,15,16,17]. In addition to using LCCs as sold with tobacco (CAI), some users smoke LCCs as ‘blunts’ (CAB) by removing the tobacco from its casing and replacing it with marijuana [18]. Using both CAI and CAB (Dual) is also common among LCC smokers [18].

To our knowledge, no studies have investigated how perceptions about COVID-19 risks affect CAI and CAB use. Concurring with the literature that suggests perceived health risk of smoking is a strong predictor of smoking cessation status or quit attempts [19,20,21], a few studies showed that a higher perceived risk of COVID-19 is related to a higher likelihood of decreasing tobacco use [8,11,22], but those findings were not specific to LCCs.

The current study aims to examine the relationship between the behavioral changes in CAI and CAB use during the first several months of the COVID-19 pandemic and important factors, including perceived risk of COVID-19 infection when smoking LCCs, user group status (i.e., CAI-only, CAB-only, or dual use), and sociodemographic characteristics (i.e., sex, age, race/ethnicity). Four behavioral changes in CAI and CAB use during the pandemic were assessed in the study: being worried about CAI/CAB use because of risk related to COVID-19, trying to quit CAI/CAB, smoking less CAI/CAB, and smoking more CAI/CAB because of COVID-19.

The current study adds to the body of cigar literature by addressing LCC use specifically as CAI-only, CAB-only, or dual use, thereby more accurately describing how young adult users smoke LCCs. As new phases of the COVID-19 pandemic continue to affect our daily lives, findings from the current study may inform future policies and cessation interventions to prevent and reduce LCC use and to effectively communicate about health risks of using LCCs to young adult LCC users.

## 2. Materials and Methods

### 2.1. Data Source and Survey Procedures

Data for the current study was collected between October and November 2020 as a part of the C’RILLOS Project, a longitudinal mixed-methods research study that investigates LCC smoking behaviors, marketing exposure, and risk perceptions among a nationally representative sample of young adults in the United States. The sample was selected from the AmeriSpeak^®^ panel of the National Opinion Research Center (NORC). According to the NORC’s report, “the AmeriSpeak^®^ panel is a probability-based panel designed to be representative of the U.S. household population” [23]. Randomly selected households were sampled using area probability and address-based sampling, with a known, non-zero probability of selection from the NORC National Sample Frame [23]. The panel provides sample coverage of approximately 97% of the U.S. household population [23].

The sample for the C’RILLOS Project was selected from the AmeriSpeak^®^ Panel using sampling strata based on age, race/Hispanic ethnicity, education, and gender. Those who were aged 18 to 34 and were LCC ever users and non-users were identified as eligible to participate in the C’RILLO Project. The online survey was administered in English and Spanish. A total of 1189 respondents completed the survey. Additional details about the C’RILLOS Project are in forthcoming publications. The current study was approved by the University of Texas Health Science Center’s Institutional Review Board.

### 2.2. Study Sample

Among those who completed the surveys, past 6 month LCC users, defined as those who had used LCCs in the past 6 months, were included in the final analytic sample (*n* = 399). The timeframe of past 6 month LCC use was selected to reflect LCC use after COVID-19 was declared a national emergency in the U.S. in March 2020.

### 2.3. Measures

#### 2.3.1. Sociodemographic Factors

Respondents self-reported age, gender (male, female), sexual orientation (heterosexual, homosexual, bisexual, other), race and ethnicity (non-Hispanic White, non-Hispanic Black, Hispanic, non-Hispanic other), educational achievement (high school or less, some college or more), and income (less than $10,000, $10,000 to $25,000, $25,000 to $50,000, $50,000 to $75,000, greater than $75,000).

#### 2.3.2. User Group Status

Respondents reported whether they had used LCCs as sold with tobacco (CAI) based on the question “Have you ever tried, even a puff or two, of a little cigar or cigarillo ‘as sold’?”, as blunts with marijuana (CAB) based on the question “Have you ever tried, even a puff or two, of a little cigar or cigarillo as a blunt?”, or dually (responding yes to both CAI and CAB questions).

Respondents were asked separately for CAI and CAB, “Which option most accurately represents the last time you used a little cigar or cigarillo [as sold/as a blunt]?” Individuals were given the response options: within the past 30 days, within the past 3 months, within the past 4 months, within the past 5 months, within the past 6 months, and more than 6 months ago. Respondents were included in the study only if they reported having used CAI or CAB within the past 6 months or less.

#### 2.3.3. Perceived Risk of Having COVID-19 when Smoking LCCs

Based on the findings from a previous focus group study in which LCC users indicated that they most often smoke a part of LCC product occasionally, [18] respondents reported their chance of having COVID-19 if they smoke a part of LCC a few days a month. This was derived from the question, “Imagine that you just began smoking a part of a little cigar or cigarillo but only a few days a month. What do you think your chances are of having the following happen to you: COVID-19 (also known as the Coronavirus)?” The responses were measured on a 5 point scale ranging from 1 (no chance) to 5 (very good chance).

#### 2.3.4. Behavioral Changes in LCC Use

Respondents were asked individual questions about changes in their CAI and/or CAB smoking behaviors because of the COVID-19 pandemic. This was measured based on the questions, “Have any of the following happened to your little cigar or cigarillo smoking [as sold/as blunt] behavior because of COVID-19?” Respondents rated their level of agreement to the following items: smoked CAI/CAB more than usual, smoked CAI/CAB less than usual, tried to stop smoking CAI/CAB, and worried about smoking CAI/CAB because of the risk associated with COVID-19. Dual users responded to both CAI and CAB-specific questions. The responses were measured on a 7 point scale ranging from 1 (strongly disagree) to 7 (strongly agree).

#### 2.3.5. Other Tobacco Use

Respondents were asked about using other tobacco products, including cigarette, large cigars, hookah, and e-cigarette (with and without marijuana) in the past 6 months prior to the survey. This was measured based on the questions, “Have you ever used a [cigarette/large cigars/hookah/e-cigarette (with and/or without marijuana)] at least once, in your lifetime? If yes, when was the most recent time you used a [cigarette/large cigars/hookah/e-cigarette (with and/or without marijuana)] even one or two puffs?” Pictures and examples of brand names of these tobacco products were shown in the survey to help respondents accurately answer the survey questions.

### 2.4. Data Analysis

All analyses were weighted to the general U.S. young adult population and were performed using SAS version 9.4 (SAS Institute, Cary, NC, USA). Chi-square tests were used to examine the differences in demographic characteristics among the user groups. Logistic regression was used to examine COVID-19 risk perception and subsequent behavioral changes specific to CAI and CAB use, controlling for demographic-level factors (age, gender, education, and race and ethnicity) and user type (e.g., CAI-Only vs. Dual). The responses for the behavioral changes in LCC use during the COVID-19 pandemic were categorized as: those who “agreed” (i.e., those who selected 5 to 7 on the scale), “disagreed” (i.e., those who selected 1 to 3 on the scale), and “neither agree nor disagree” (i.e., those who selected 4 on the scale) to experiencing changes in LCC use behaviors. Those who neither agreed nor disagreed were excluded from the regression analysis.

## 3. Results

### 3.1. Demographic Characteristics

Among 399 past 6 month LCC users in our sample, 72 were CAI-only users (14.5%), 160 were CAB-only users (38.0%), and 167 were dual users (47.4%) (Table 1). The mean age of our sample was 26.9 (SD 4.7). Overall, respondents were 56.7% male, 77.3% heterosexual, 43.4% non-Hispanic White, and 55.0% had some college or more education. The mean perceived risk of having COVID-19 when smoking LCCs was 2.6 (SD 1.2), for all respondents with no difference among the user groups. Significant differences among the user groups were observed for gender, sexual orientation, race and ethnicity, and educational achievement (*p* < 0.01).

Past 6 month CAI-only users were 68.4% male, 86.8% heterosexual, 42.4% Hispanic, and 73.9% had some college experience. Past 6 month CAB-only users were 56.4% female, 70.4% heterosexual, 43.7% non-Hispanic White, and 63.4% had some college experience. Past 6 month dual users were 63.6% male, 79.9% heterosexual, 35.7% non-Hispanic White, and 57.6% had high school or less education.

Consistent with previous studies [24,25], poly use of tobacco products was common in our sample. In addition to smoking LCCs, 66.0% used cigarettes, 61.1% used large cigars, 60.0% used e-cigarettes without marijuana, 76.4% used e-cigarettes with marijuana, and 37.0% used hookah in the past 6 months prior to the survey. There were differences between user groups in their concurrent use of large cigars, e-cigarettes without marijuana, and hookah (*p* < 0.01).

### 3.2. Behavioral Changes in LCC Use during the COVID-19 Pandemic

Overall, respondents did not report experiencing substantial changes in CAI and CAB use during the first several months of the COVID-19 pandemic (Table 2). Among CAI-only users, 12.6% agreed with smoking more CAI during the pandemic while 16.9% agreed with smoking less CAI. Among CAB-only users, only 6% agreed with trying to quit CAB, and 14.5% agreed with smoking less CAB. Also, only 7.7% of dual users agreed with smoking less CAB during the pandemic. There were no differences observed between CAI-only and dual users regarding the behavioral changes in CAI use during the pandemic. There were differences (*p* < 0.01) between CAB-only and dual users regarding worrying about smoking CAB, trying to quit CAB, and smoking less CAB during the pandemic.

### 3.3. Factors Associated with the Behavioral Changes in LCC Use during the COVID-19 Pandemic

#### 3.3.1. Changes in Using LCCs as Sold with Tobacco (CAI Use)

Compared to the non-Hispanic White population, the non-Hispanic Black population were more likely to endorse worrying about CAI use because of the risk associated with COVID-19 (OR = 3.14, 95% CI = 1.17, 8.45), controlling for other sociodemographic characteristics and risk perceptions (Table 3). Those with a greater perceived risk of having COVID-19 if using LCCs were more likely to endorse trying to quit CAI (OR = 1.56, 95% CI = 1.16, 2.09) during the COVID-19 pandemic. On the other hand, older LCC users were less likely to endorse trying to quit CAI during the pandemic compared to younger LCC users (OR = 0.89, 95% CI = 0.82, 0.96). In addition, compared to the non-Hispanic White population, non-Hispanic Others were more likely to endorse trying to quit CAI (OR = 5.96, 95% CI = 1.08, 33.00). Compared to males, females were less likely to endorse smoking less CAI during the pandemic (OR = 0.40, 95% CI = 0.17, 0.95). Also, compared to the non-Hispanic White population, the non-Hispanic Black population were less likely to endorse smoking less CAI (OR = 0.18, 95%CI = 0.05, 0.62) during the pandemic. Users with some college or more education were more likely to endorse smoking less CAI compared to those with high school or less education (OR = 3.11, 95% CI = 1.25, 7.75).

#### 3.3.2. Changes in Using LCCs with Marijuana as Blunts (CAB Use)

Compared to CAB-only users, dual users were less likely to endorse worrying about CAB use because of the risk related to COVID-19 (OR = 0.46, 95% CI = 0.25, 0.87) and smoking less CAB during the pandemic (OR = 0.38, 95% CI = 0.17, 0.82) after controlling for other characteristics. They were also more likely to endorse smoking more CAB (OR = 2.15, 95% CI = 1.21, 3.82) than CAB-only users. Those with greater perceived risk of having COVID-19 if using LCCs were more likely to endorse worrying about CAB (OR = 1.31, 95% CI = 1.03, 1.67) and trying to quit CAB (OR = 1.36, 95% CI = 1.01, 1.84) during the pandemic. The non-Hispanic Black population were less likely to endorse trying to quit CAB (OR = 0.26, 95% CI = 0.08, 0.87) compared to non-Hispanic White population. Moreover, compared to males, females were more likely to endorse smoking more CAB (OR = 2.27, 95% CI = 1.28, 4.01) during the pandemic.

## 4. Discussion

The current study examined factors that are associated with the reported behavioral changes in LCC use during the first several months of the COVID-19 pandemic among young adult users, specified by the modality of use (i.e., LCC use with tobacco (CAI), LCC use with marijuana as blunts (CAB) and user group status (i.e., dual use, exclusive use). Descriptive analysis of our sample indicated that dual use of CAI and CAB was the most prevalent (47.4%), compared to exclusive CAB (38.0%) and CAI use (14.5%). Such a finding is in line with emerging studies that reported increased sales of cannabis in some U.S. states [26] and increased use of marijuana across sub-populations during the pandemic [27,28]. The movement toward the legalization of recreational and medical marijuana may account for the increasing prevalence of blunt use among young adults [27]. Reasons for marijuana use during the pandemic overlap with the reasons for tobacco use during the pandemic, including self-isolation, more free time, and coping with stress and depression [27,29].

Respondents in our sample did not report drastic changes in LCC use during the pandemic. Still, we observed racial and ethnic differences in CAI and CAB use behaviors. Compared to the non-Hispanic White population, the non-Hispanic Black population were less likely to endorse trying to quit CAB and smoking less CAI during the pandemic, although they were more likely to endorse worrying about smoking CAI because of risk related to COVID-19. The heightened concern about smoking CAI may be due to the disproportionate impact of COVID-19 on communities of color [30]. Yet, their concern did not lead to smoking less CAI, because smoking may have been their primary coping mechanism to manage their negative affect. Further exploration is needed to understand the discrepancy between worrying about CAI use and smoking less CAI among the non-Hispanic Black population.

In addition, females were less likely to endorse smoking less CAI and more likely to endorse smoking more CAB than males during the pandemic. One likely explanation for the observed racial/ethical and gender differences is that pandemic-induced stressors, such as financial concerns, death of loved ones, feeling isolated [31,32], and the responsibility of parenting and elder care [33], have disproportionately affected specific populations, including communities of color and women [33,34]. COVID-19 has disproportionately affected communities of color [30], and women have been shown to experience greater degrees of stressfulness than men over the course of the pandemic [31,33,35]. This highlights the need for greater mental health support to help LCC smokers better manage pandemic-induced stress and dissuade them from smoking to reduce stress.

We also found that user type (i.e., CAI-only, CAB-only, dual users) had an impact on COVID-related behavioral changes in LCC use. Compared to CAB-only users, dual users were more likely to endorse smoking more CAB and less likely to endorse smoking less CAB during the pandemic. Because dual users use multiple tobacco products and consume both tobacco and marijuana, they may experience greater nicotine dependence [36,37,38] than CAB-only users. Additional research on the patterns and reasons for CAB use is needed to understand the differences between CAB-only users and dual users.

Our findings showed that users with a higher perceived risk of having COVID-19 when smoking LCCs were more likely to endorse trying to quit both CAI and CAB during the pandemic. Nevertheless, the perceived risk of having COVID-19 when smoking LCCs was low in our sample. This finding builds upon previous research that reported cigar smokers, including LCC users, tend to underestimate its health risks compared to cigarettes [22,39,40]. Moreover, perceived COVID-19 risk may be related to how LCCs are smoked. For instance, young adults who smoke LCCs alone may feel their risk of having COVID-19 is low compared to those who smoke LCCs in social settings in close proximity to others and share the inhaled LCCs.

Preliminary research shows that tobacco use is associated with more severe symptoms and progression of COVID-19 [41,42]. In addition, increased CAI and CAB use during the pandemic can lead to sustained nicotine and marijuana dependences and addiction, increasing risks of developing subsequent tobacco- and marijuana-related diseases, such as lung cancer, heart disease, stroke, emphysema, and chronic obstructive pulmonary disease (COPD). Thus, interventions seeking to reduce LCC use among young adults should raise their awareness about the specific health risks of smoking LCCs and its possible effects on COVID-19 outcomes.

This study has a few limitations. The behavioral changes in LCC use during the COVID-19 pandemic were self-reported and were not directly measured using biological measures. There is also a potential recall bias in reporting past LCC use. Moreover, while this study is representative of the young adult population in the U.S., it is important to note the limited sample size included in the logistic regression analysis, particularly for the CAI-only users.

## 5. Conclusions

We examined young adult LCC users’ perceived risk of COVID-19 and their behavioral changes in LCC use in response to the COVID-19 pandemic. Our findings showed that young adult LCC users, particularly the non-Hispanic Black population, dual users, and females, are at higher risk of increased LCC use during the pandemic. Smoking behaviors formed during the first few waves of the COVID-19 pandemic, as well as during the repeated outbreaks caused by new variants and subvariants of the virus, will continue to affect the health of LCC users. Furthermore, because the Food and Drug Administration (FDA) is considering stricter regulations on tobacco products, such as a JUUL e-cigarette ban and reducing allowable nicotine levels in cigarettes, it is possible that users of these tobacco products switch to using LCCs. Therefore, our findings may inform the development of tobacco prevention and cessation interventions. Programs seeking to reduce LCC use must have focused efforts to reduce all modalities of LCC use, including LCC use with tobacco (CAI) and with marijuana as blunts (CAB), among vulnerable sub-populations of young adult LCC users.

## Figures and Tables

**Table 1 ijerph-19-08933-t001:** Demographic characteristics of past 6 month LCC users by use modality.

	Any LCC Use ^1^	Only Use LCCs with Tobacco (CAI-Only)	Only Use LCCs as Blunts (CAB-Only)	Dual Use ^1^	*p*-Value
	*n* = 399	*n* = 72 (14.5%)	*n* = 160 (38.0%)	*n* = 167 (47.4%)	
**Age Category, mean (SD)**	26.9 (4.7)	28.1 (4.1)	26.1 (4.2)	27.1 (5.3)	0.81
**Gender, %**					<0.01
Male	56.7	68.4	43.6	63.6	
Female	43.3	31.6	56.4	36.4	
**Sexual Orientation, %**					<0.01
Heterosexual/Straight	77.3	86.8	70.4	79.9	
Homosexual/Gay or Lesbian	5.2	1.2	10.0	2.6	
Bisexual	13.5	8.7	15.4	15.2	
Other	2.0	3.0	4.1	0.1	
**Race & Ethnicity, %**					<0.01
Non-Hispanic White	43.4	1.9	43.7	35.7	
Non-Hispanic Black	22.5	24.3	23.1	24.6	
Hispanic	26.6	42.4	24.6	32.5	
Non-Hispanic Other	7.4	31.5	8.6	7.3	
**Educational Achievement, %**					<0.01
High school or less	45.0	26.1	36.6	57.6	
Some college or more	55.0	73.9	63.4	42.4	
**Income, %**					0.26
Less than $10,000	10.1	5.3	13.6	10.2	
$10,000 to $25,000	15.3	20.6	17.3	12.0	
$25,000 to $50,000	26.1	23.4	21.4	30.7	
$50,000 to $75,000	19.6	17.1	19.7	20.2	
Greater than $75,000	28.3	33.6	28.0	26.9	
**Other Tobacco Use ** **in the past 6 months, %**					
Cigarette	66.0	64.2	67.2	65.7	0.94
Large Cigar	61.1	66.9	30.1	77.7	<0.01
E-Cigarette without marijuana	60.0	67.9	45.1	71.0	<0.01
E-Cigarette with marijuana	76.4	66.0	75.8	79.6	0.42
Hookah	37.0	25.8	16.2	61.7	<0.01
**Perceived COVID-19 Risk ^2^, ** **mean (SD)**	2.6 (1.2)	2.4 (1.2)	2.5 (1.2)	2.8 (1.2)	0.16

^1^ Any LCC use: Includes using LCCs as sold with tobacco (CAI), LCCs with marijuana as blunts (CAB), and/or dual use; Dual use: Includes using LCCs with tobacco (CAI) and as blunts (CAB). ^2^ Perceived risk of having COVID-19 when smoking LCCs was measured on a 5 point scale ranging from 1 (no chance) to 5 (very good chance). *All percentages are weighted to general U.S. population.

**Table 2 ijerph-19-08933-t002:** Behavioral changes in LCC use during COVID-19 pandemic among past 6 month users.

Behavior Changes in Using LCCs with Tobacco (CAI)	Only Using LCCs with Tobacco (CAI-Only) *	Dual Use *	*p*-Value
	Unweighted *n* = 71	Unweighted *n* = 163	
**Worried About CAI use ** **because of risks associated with COVID-19**			0.21
Disagree	46.6	34.5	
Neither Agree nor Disagree	31.7	35.6	
Agree	21.7	29.9	
**Tried to Quit CAI**			0.09
Disagree	40.3	35.0	
Neither Agree nor Disagree	39.7	30.3	
Agree	20.0	34.7	
**Smoked Less CAI**			0.49
Disagree	44.3	46.2	
Neither Agree nor Disagree	38.8	31.5	
Agree	16.9	22.3	
**Smoked More CAI**			0.09
Disagree	50.9	38.4	
Neither Agree nor Disagree	36.5	37.3	
Agree	12.6	24.3	
**Behavior Changes in ** **Using LCCs as Blunts (CAB)**	**Only Using LCCs ** **as Blunts ** **(CAB-Only) ***	**Dual Use ***	***p*-Value**
	**Unweighted *n* = 159**	**Unweighted *n* = 163**	
**Worried About CAB use ** **because of risks associated with COVID-19**			<0.01
Disagree	38.2	50.3	
Neither Agree nor Disagree	33.0	35.4	
Agree	28.9	14.3	
**Tried to Quit CAB**			<0.01
Disagree	53.4	58.1	
Neither Agree nor Disagree	40.6	28.1	
Agree	6.0	13.8	
**Smoked Less CAB**			<0.01
Disagree	42.2	65.3	
Neither Agree nor Disagree	43.4	27.0	
Agree	14.5	7.7	
**Smoked More CAB**			0.09
Disagree	38.7	28.1	
Neither Agree nor Disagree	28.9	30.8	
Agree	32.4	41.1	

* CAI-only: Only using LCCs as sold with tobacco; CAB-only: Only using LCCs with marijuana as blunts; Dual use: Using LCC both as CAI and CAB.

**Table 3 ijerph-19-08933-t003:** Factors associated with behavioral changes in LCC use among past 6 month users.

	Using LCCs with Tobacco (CAI)			Using LCCs as Blunts (CAB)		
	OR	95% CI		OR	95% CI	
**Endorsing to Worrying about LCC Use because of COVID-19 Risk**	*n* = 140			*n* = 222		
Dual Product User ^1^	1.27	0.82	1.96	**0.46**	**0.25**	**0.87**
Risk Perception ^2^	1.16	0.88	1.53	**1.31**	**1.03**	**1.67**
Age ^2^	0.96	0.89	1.04	0.94	0.87	1.01
Female ^3^	0.54	0.26	1.13	0.99	0.53	1.84
Non-Hispanic Black ^4^	**3.14**	**1.17**	**8.45**	1.33	0.62	2.86
Hispanic ^4^	1.02	0.42	2.48	0.93	0.45	1.95
Non-Hispanic Other ^4^	1.13	0.22	5.73	0.73	0.20	2.77
Some College or More ^5^	1.53	0.69	3.41	1.40	0.74	2.62
**Endorsing to Trying to Quit LCCs**	*n* = 139			*n* = 217		
Dual Product User ^1^	1.00	0.63	1.59	2.12	0.89	5.04
Risk Perception ^2^	**1.56**	**1.16**	**2.09**	**1.36**	**1.01**	**1.84**
Age ^2^	**0.89**	**0.82**	**0.96**	1.07	0.99	1.16
Female ^3^	1.38	0.66	2.89	1.60	0.74	3.44
Non-Hispanic Black ^4^	1.10	0.40	2.97	**0.26**	**0.08**	**0.87**
Hispanic ^4^	2.30	0.94	5.64	0.75	0.32	1.75
Non-Hispanic Other ^4^	**5.96**	**1.08**	**33.00**	0.42	0.05	3.64
Some College or More ^5^	0.77	0.35	1.70	1.20	0.55	2.65
**Endorsing to Smoking Less LCCs**	*n* = 135			*n* = 210		
Dual Product User ^1^	1.54	0.90	2.64	**0.38**	**0.17**	**0.82**
Risk Perception ^2^	1.34	0.97	1.87	1.07	0.79	1.45
Age ^2^	1.09	0.99	1.19	0.98	0.90	1.07
Female ^3^	**0.40**	**0.17**	**0.95**	0.99	0.46	2.10
Non-Hispanic Black ^4^	**0.18**	**0.05**	**0.62**	0.40	0.14	1.17
Hispanic ^4^	0.39	0.15	1.04	1.04	0.45	2.41
Non-Hispanic Other ^4^	0.58	0.13	2.63	1.79	0.45	7.06
Some College or More ^5^	**3.11**	**1.25**	**7.75**	1.67	0.77	3.60
**Endorsing to Smoking More LCCs**	*n* = 141			*n* = 218		
Dual Product User ^1^	1.41	0.88	2.25	**2.15**	**1.21**	**3.82**
Risk Perception ^2^	0.98	0.74	1.30	0.94	0.75	1.17
Age ^2^	1.07	0.98	1.16	0.96	0.90	1.02
Female ^3^	1.62	0.78	3.38	**2.27**	**1.28**	**4.01**
Non-Hispanic Black ^4^	2.12	0.78	5.75	0.97	0.48	1.96
Hispanic ^4^	1.56	0.65	3.75	1.07	0.55	2.09
Non-Hispanic Other ^4^	1.40	0.30	6.53	0.70	0.20	2.40
Some College or More ^5^	1.37	0.63	2.98	0.78	0.45	1.38

Referent: ^1^ Single product user (e.g., CAI-only, CAB-only) is the referent compared to dual product user; ^2^ Risk perception and age were continuous variables; ^3^ Males are the referent compared to females; ^4^ Non-Hispanic White is the referent for all race and ethnic groups; ^5^ High school or less is the referent.

## Data Availability

The data presented in this study are available on request from the corresponding author.

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
