# Peer review of "Changes in Little Cigar and Cigarillo Use during the COVID-19 Pandemic: A Cross-Sectional Analysis of a Nationally Representative Sample of Young Adult Users"

_ijerph, 2022, doi:10.3390/ijerph19158933_

Round 1
Reviewer 1 Report
Excellent paper! Important topic and well written. That said, I have a few suggestions.
I find the acronyms CAI and CAB to be not at all intuitive. I recommend finding something that's easier for the reader to remember.
In some places (e.g. pp. 6, 8), I recommend changing word choices for words like less/more when they are used in different contexts within the same sentence. For example you discuss behavior being less/more likely, and the behavior involves smoking less/more. If you're tracking the language in the questions, I'd understand if you're not comfortable adopting this edit. It's just a suggestion.
On p.8 you say dual users are more likely to experience nicotine dependence and marijuana dependence than CAB-only smokers. I understand why they'd be more likely to experience nicotine dependence. But it's unclear why they're more likely to experience marijuana dependence. I think you need to explain why. It seems counterintuitive.
Author Response
Comment 1: I find the acronyms CAI and CAB to be not at all intuitive. I recommend finding something that's easier for the reader to remember.
Response to Comment 1: Thank you for your suggestion. We acknowledge that the abbreviations are not intuitive. To help the readers, we will write out the full product names in the data tables and the first time they appear in each section of the manuscript (i.e., Introduction, Materials and Methods, Discussion, Conclusion). However, in the results section, we will use abbreviations because they improve the readability of the numerical results.
Comment 2: In some places (e.g. pp. 6, 8), I recommend changing word choices for words like less/more when they are used in different contexts within the same sentence. For example you discuss behavior being less/more likely, and the behavior involves smoking less/more. If you're tracking the language in the questions, I'd understand if you're not comfortable adopting this edit. It's just a suggestion.
Response to Comment 2: Thank you for your comment. As you mentioned, we are tracking the language in the questions. Therefore, no changes have been made.
Comment 3: On p.8 you say dual users are more likely to experience nicotine dependence and marijuana dependence than CAB-only smokers. I understand why they'd be more likely to experience nicotine dependence. But it's unclear why they're more likely to experience marijuana dependence. I think you need to explain why. It seems counterintuitive.
Response to Comment 3: Thank you for your response. After careful evaluation of your comment, we agree with your feedback. Therefore, the phrase is now removed from the sentence.
Reviewer 2 Report
This is an interesting study on a significant public health problem.
Please consider the following changes to clarify the manuscript.
1. Methods section - please provide a reference or more precise data on sampling/data collection methods - e.g. -> " The sample was selected 83 from the AmeriSpeak® panel of the National Opinion Research Center (NORC) []". The major issue is - how this method guarantees the representativeness of the study sample?
2. "CAI" and "CAB" abbreviations seems not to be standard abbreviation. To clarify the data interpretation please consider the full names of the products in the leading (first) subheading of the table. It will ease the reading of the text.
3. Please consider adding 2-3 sentences on the practical implications of this study and further research needs.
This is a really good and important study, congratulations!
Author Response
Comment 1. Methods section - please provide a reference or more precise data on sampling/data collection methods - e.g. -> " The sample was selected 83 from the AmeriSpeak® panel of the National Opinion Research Center (NORC) []". The major issue is - how this method guarantees the representativeness of the study sample?
Response to Comment 1: Thank you for your response. Per your suggestion, more information on sampling is now added to the manuscript. Also, more detailed information on panel methodology is available upon request.
2. "CAI" and "CAB" abbreviations seems not to be standard abbreviation. To clarify the data interpretation please consider the full names of the products in the leading (first) subheading of the table. It will ease the reading of the text.
Response to Comment 2: Thank you for your suggestion. We acknowledge that they are not intuitive. We will write out the full product names in the data tables and the first time they appear in each section of the manuscript (i.e., Introduction, Materials and Methods, Discussion, Conclusion). However, in the results section, we will use abbreviations because they improve the readability of the numerical results.
3. Please consider adding 2-3 sentences on the practical implications of this study and further research needs.
Response to Comment 3: Thank you for your response. We will add a few sentences on the practical implications and further research needs per your suggestion.
Reviewer 3 Report
The authors describe changes in little cigar and cigarillo use by young adults during the COVID-19 pandemic. Their manuscript adds to the literature by addressing little cigar/cigarillo use as tobacco only, marijuana only, and dual use; thereby the authors more accurately describing how young adults smoke LCCs. This work is especially timely as the FDA considers reducing allowable nicotine levels in cigarettes, but not in little cigars and cigarillos. Thus, cigarette smokers may migrate to LCCs in the future and thus more information is needed about LCC use and health risk perceptions.
The following comments are intended to strengthen the manuscript:
1. User groups would be strengthened by including a daily use group for LCCs. Currently the most frequent use group could be 1/30 days. It would also be valuable to understand the primary product used by poly-users.
2. Asking about sources for health information would have strengthened the study (e.g. what sources of information influenced your perceptions around smoking and COVID risk?)
3. Table 2 formatting is confusing, especially at page break.
4. Semantics: “perceived risk of having COVID-19 from LCC use”. Obviously LCC use doesn’t cause COVID-19; rather, smoking increases the risk of contracting COVID-19 and the risk that that COVID infection is more serious. Please update your phrasing appropriately.
5. Did all study subjects live in areas with legalized non-medical cannabis use? If not, what was the distribution of legal/illegal locales and did that impact health perceptions?
6. How were “quit attempts” defined? Was it just quitting LCCs or was it quitting all tobacco products?
7. The “dose makes the poison”; did you assess whether risk perception was related to LLC smoking frequency?
8. Endorsement is somewhat spongy and subjective. Some kind of biological measure of smoking would have strengthened the study (e.g. exhaled CO, serum cotinine, urinary cannabinoids, etc.)
Author Response
Please see the attachment.
Thank you so much for your feedback on the manuscript.
